# Untargeted Metabolomics and Antioxidant Capacities of Muscadine Grape Genotypes during Berry Development

**DOI:** 10.3390/antiox10060914

**Published:** 2021-06-04

**Authors:** Ahmed G. Darwish, Protiva Rani Das, Ahmed Ismail, Pranavkumar Gajjar, Subramani Paranthaman Balasubramani, Mehboob B. Sheikh, Violeta Tsolova, Sherif M. Sherif, Islam El-Sharkawy

**Affiliations:** 1Center for Viticulture and Small Fruit Research, College of Agriculture and Food Sciences, Florida A&M University, Tallahassee, FL 32308, USA; ahmed.darwish@famu.edu (A.G.D.); provita@vt.edu (P.R.D.); ahmed.ismail@agr.dmu.edu.eg (A.I.); pranavkumar1.gajjar@famu.edu (P.G.); balasubramani.subramaniparanthaman@asurams.edu (S.P.B.); mehboob.sheikh@famu.edu (M.B.S.); violeta.tsolova@famu.edu (V.T.); 2Department of Biochemistry, Faculty of Agriculture, Minia University, Minia 61519, Egypt; 3Alson H. Smith Jr. Agricultural Research and Extension Center, School of Plant and Environmental Sciences, Virginia Tech, Winchester, VA 22602, USA; ssherif@vt.edu; 4Department of Horticulture, Faculty of Agriculture, Damanhour University, Damanhour 22516, Egypt; 5Department of Natural Sciences, Albany State University, Albany, GA 31707, USA

**Keywords:** developmental stages, metabolomics, muscadine genotypes, nutritional biomarkers

## Abstract

Three muscadine grape genotypes (*Muscadinia rotundifolia* (Michx.) Small) were evaluated for their metabolite profiling and antioxidant activities at different berry developmental stages. A total of 329 metabolites were identified using UPLC-TOF-MS analysis (Ultimate 3000LC combined with Q Exactive MS and screened with ESI-MS) in muscadine genotypes throughout different developmental stages. Untargeted metabolomics study revealed the dominant chemical groups as amino acids, organic acids, sugars, and phenolics. Principal component analysis indicated that developmental stages rather than genotypes could explain the variations among the metabolic profiles of muscadine berries. For instance, catechin, epicatechin-3-gallate, and gallic acid were more accumulated in ripening seeds (RIP-S). However, tartaric acid and malonic acid were more abundant during the fruit-set (FS) stage, and malic acid was more abundant in the veraison (V) stage. The variable importance in the projection (VIP > 0.5) in partial least-squares–discriminant analysis described 27 biomarker compounds, representing the muscadine berry metabolome profiles. A heatmap of Pearson’s correlation analysis between the 27 biomarker compounds and antioxidant activities was able to identify nine antioxidant determinants; among them, gallic acid, 4-acetamidobutanoic acid, trehalose, catechine, and epicatechin-3-gallate displayed the highest correlations with different types of antioxidant activities. For instance, DPPH and FRAP conferred a similar antioxidant activity pattern and were highly correlated with gallic acid and 4-acetamidobutanoic acid. This comprehensive study of the metabolomics and antioxidant activities of muscadine berries at different developmental stages is of great reference value for the plant, food, pharmaceutical, and nutraceutical sectors.

## 1. Introduction

Muscadine grapes are well-known grape species native to the southeastern region of the United States and were first cultivated in North America more than 400 years ago [1,2]. Due to their chemical compositions and nutritional benefits, muscadine grapes attract significant attention from the food, pharmaceutical, and nutraceutical sectors. Several studies demonstrated that bioactive constituents from muscadine have potent antioxidant, anticancer, antimutagenic, antimicrobial, and anti-inflammatory properties [3,4,5,6]. Muscadine constituents have also been reported to possess cardio-protective activities and defend against colon dysbiosis [7,8]. Accordingly, muscadine and its products can be exploited for their nutraceutical qualities and health-promoting properties. Muscadine grape contains unique sets of primary and secondary metabolites, including fruit acids, carbohydrates, and phenolics, especially gallic acid, ellagic acid, proanthocyanidins, catechins, quercetin, resveratrol, and myricetin [9,10,11]. In addition to their health-promoting properties, grape metabolites also play crucial roles in the quality parameters and broad taste spectrum of grapes [12,13,14].

The daily consumption of bioactive metabolites from plant-based food sources is considered a more practical preventive approach against several health complications than taking dietary supplements and synthetic products [15]. Consequently, there is a growing trend for promoting functional food efficacy via stimulating the accumulation of naturally occurring metabolites in its natural biological matrix. To this end, several factors should be taken into consideration, including, but not necessarily limited to, genotypic variations of the same species, developmental stages at which bioactive compounds are displayed, environmental cues regulating the abundance of target metabolites, and the interactions among these factors. Among developmental factors, berry phenology has a crucial role in determining the accumulation kinetics of various bioactive compounds, which, in addition to their health benefits, are intrinsically associated with the changes in the berry’s color, texture, and flavor through development and maturity [14,16]. In grapes, berry development is typically divided into stage I (berry formation to lag phase), stage II (lag phase to veraison), and stage III (post-veraison to ripening) [17].

The available research on muscadine grapes mainly covers the extraction procedures, metabolic profiles among berry parts, cultivar variations, nutraceuticals, and potential health properties of berries [2,18]. However, little is known about the changes in the types and levels of various metabolites during berry developmental stages. Recently, high-throughput metabolomics technologies have allowed real-time and unbiased identification and quantification of numerous metabolites, paving the way for several biological applications [14,19,20,21].

To the best of our knowledge, the untargeted metabolomics and antioxidant activities of muscadine genotypes at different berry developmental stages have never been evaluated. Therefore, the current study aimed to explore the metabolic profiles of three muscadine genotypes throughout six developmental stages to provide a comprehensive database of the types, quantities, and accumulation kinetics of biologically active metabolites that could be targeted by breeding programs for enhancing the nutritional value of existing muscadine germplasm. A multivariate study was also performed to characterize biomarker compounds with particular antioxidant potential, which could be of great importance for the food, pharmaceutical, and nutraceutical sectors.

## 2. Materials and Methods

### 2.1. Chemicals

Folin–Ciocalteu phenol reagent, 2,2-diphenyl-1-picrylhydrazyl (DPPH), gallic acid, quercetin, Trolox, HPLC-grade methanol, acetic acid, quercetin, 2,4,6-tripyridyl-s-triazine (TPTZ), 2,2′-azinobis-(3-ethylbenzothiazoline-6-sulfonic acid) (ABTS), sodium nitroprusside (SNP), phenazine methosulfate (PMS), sulfanilamide, naphthyl ethylenediamine dihydrochloride (NED), neocuproine, glacial acetic acid, sodium acetate trihydrate, and ferric chloride (FeCl_3_) were all purchased from Sigma (Sigma-Aldrich, St. Louis, MO, USA).

### 2.2. Muscadine Grape Materials

Three muscadine grape genotypes (*Muscadinia rotundifolia* (Michx.) Small), i.e., one standard cultivar Late Fry (LF) (Plant patent #9224) as well as two breeding lines C5-9-1 (C5) and C6-10-1(C6), were used in this study. All muscadine genotypes were grown at the Center for Viticulture, Tallahassee, Florida (30°28′45.63′′ N, 84°10′16.43′′ W). Vineyard management and practices followed the guidelines outlined in the *Muscadine Production Guide for Florida* written by the Center for Viticulture and Small Fruit Research (CVSFR), Florida Agricultural & Mechanical University (FAMU) (http//famu.edu/viticulture). The C5 and C6 breeding lines were developed under the grape-breeding program of the Center for Viticulture and Small Fruit Research, Florida Agricultural & Mechanical University (Tallahassee, FL, USA). The three genotypes were selected according to the diversity in their total phenolic content and total flavonoid content, as well as DPPH radical-scavenging activity at ripening [6]. Samples were collected from 5-year-old grapevines at different developmental stages: fruit-set (FS), pre-veraison (pre-V), veraison (V), post-veraison (post-V), and ripening (RIP) stages. The berries were carefully separated into two different tissues at the ripening stage, designated as ripe skin/flesh (RIP-SF) and ripen seeds (RIP-S). Five clusters/replicate and three replicates/sample were randomly collected for all developmental stages, except for the FS stage. At FS, 70 clusters/replicate were collected due to the small berry size. All samples were immediately frozen in liquid nitrogen and stored at −80 °C for further analysis.

### 2.3. Preparation of Muscadine Extracts

All frozen samples were ground to a fine powder using a Geno/Grinder 2010 (Metuchen, NJ, USA). Next, 12 grams of sample powder were subjected to methanol extraction using 100 mL of methanol. All extractions were performed under shaking (150 rpm) for 24 h in the dark at room temperature. Then, the extracts were filtered through Whatman No. 41 filter papers (Thomas Scientific, Swedesboro, NJ, USA). The collected supernatant was concentrated using a Heidolph rotary evaporator (Thermo Fisher Scientific, Waltham, MA, USA) at 40 °C and then dehydrated using a speed vacuum (Eppendorf, Enfield, CT, USA). All dried extracts were stored at 4 °C in the dark for further analysis. The stock solution of grape extracts was prepared at 10 mg/mL concentration in DMSO to determine the total metabolite content and antioxidant activities.

### 2.4. Analysis of Total Phenolic and Flavonoid Contents

Total phenolic content (TPC) was assessed according to the Folin–Ciocalteu colorimetric method based on the previously described protocol with minor modifications [22]. Briefly, 15 μL of diluted samples were placed in a 96-well microplate (Genesee Scientific, San Diego, CA, USA). Subsequently, 240 μL of Folin–Ciocalteu reagent (1:15, *v*/*v*) was added to the wells, mixed with samples, and incubated in the dark at room temperature for 30 min. Then, the mixtures were treated with 15 μL of 20% sodium carbonate solution. The mixture was shaken before measuring the absorbance at λ = 755 nm using a microplate reader (ACCURIS SmartReader; Edison, NJ, USA). Gallic acid solutions in the specific concentration range were used to construct a calibration curve. TPC estimation was performed for each biological replicate in triplicate (*n* = 9) and expressed as milligram gallic acid equivalents per gram of sample fresh weight (mg GAE/g FW).

Total flavonoid content (TFC) was estimated based on the previously reported method with slight modifications [23]. Briefly, an aliquot (25 μL) of diluted samples was mixed with 75 μL of 96% methanol (*v*/*v*) and placed in a 96-well microplate. Then, 5 μL of 10% aluminum chloride and 5 μL of potassium acetate (1M) were added to the mixture. Finally, 140 μL of distilled water was added to the mix and incubated for 30 min in the dark at room temperature. The mixture was shaken before measuring the absorbance at λ = 415 nm using a microplate reader. Quercetin was used to construct the calibration curve in a different concentration range. Total flavonoid contents were estimated for each biological replicate in triplicate (*n* = 9) and expressed as milligram quercetin equivalents per gram of sample fresh weight (mg QE/g FW).

### 2.5. Analysis of Antioxidant Activities

#### 2.5.1. DPPH Radical-Scavenging Activity

DPPH radical-scavenging activity was assayed according to the previously reported method [24]. Briefly, 100 μL of diluted samples was mixed with 100 μL of freshly prepared DPPH methanolic solution (200 μM). The mixture was then incubated for 30 min in the dark at room temperature. The absorbance was measured at λ = 515 nm using a microplate reader. As a control, DMSO was used in place of the muscadine samples. DPPH-scavenging activity was estimated for each biological replicate in triplicate (*n* = 9). Data were expressed as the percentage scavenging of DPPH radicals and calculated using the following equation:DPPH (%) = [1 − (A_sample_ − A_background_)/(A_DMSO_ − A_background_)] × 100

#### 2.5.2. Ferric Reducing Antioxidant Potential (FRAP) Assay

The FRAP assay was performed based on the previously described method [25]. FRAP reagent was prepared at the ratio of 10:1:1 (*v/v/v*) comprising 300 mM acetate buffer (pH 3.6), a solution of 40 mM TPTZ in 40 mM HCl, and 20 mM FeCl_3_. Freshly prepared FRAP reagent (280 µL) and diluted samples (20 µL) were mixed in a 96-well microplate and incubated at 37 °C in the dark for 30 min. The absorbance was measured at λ = 590 nm using the microplate reader. Trolox at different concentrations was used to make the standard curve. FRAP radical-scavenging activity was estimated for each biological replicate in triplicate (*n* = 9). Data are expressed in micro-molar Trolox equivalents per gram of sample fresh weight (µM TE/g FW).

#### 2.5.3. ABTS Assay

ABTS assay was performed based on the previously described method with minor modifications [26]. Briefly, a stock solution was prepared by mixing ABTS solution (7 mM) and potassium persulfate solution (2.4 mM) in equivalent amounts. The mixture was incubated to react in the dark at room temperature for 14 h. Then, methanol was added to the resulting mixture for dilution to get the absorbance as 0.7 ± 0.01 units at λ = 734 nm. After that, 1 mL of each diluted sample was allowed to react with 1 mL of freshly prepared ABTS solution, and the absorbance was measured after 7 min at λ = 734 nm. The activity was estimated for each biological replicate in triplicate (*n* = 9). ABTS radical-scavenging activity was calculated based on the following equation:ABTS (%) = ((A_control_ − A_sample_)/A_control_) × 100
where A_control_ means absorbance of ABTS radical + methanol and A_sample_ means absorbance of ABTS radical + sample.

#### 2.5.4. Cupric Ion Reducing Antioxidant Capacity (CUPRAC) Assay

The CUPRAC assay was performed according to the previously described method [27]. Briefly, 100 μL of the sample was mixed with 1 mL of copper chloride solution (10 mM), neocuproine alcoholic solution (7.5 mM in absolute ethanol), ammonium acetate buffer solution (1 M, pH 7.0), and distilled water (final volume 4.1 mL). After incubation for 30 min, the absorbance was recorded at λ = 450 nm against the reagent blank. Different concentrations of Trolox were used to generate the standard curve. CUPRAC activity was estimated for each biological replicate in triplicate (*n* = 9). The data were expressed as micro-molar Trolox equivalents per gram of sample fresh weight (μmol TE/g FW).

#### 2.5.5. Nitric Oxide Radical-Scavenging (NORS) Assay

The NORS assay was performed according to the previously proposed method [28]. In this assay, 3 mL of reaction mixture containing 10 mM aqueous sodium nitroprusside (SNP) solution (in PBS, pH 7.4) was mixed with diluted samples. The mixture was incubated at 25 °C for 150 min; after that, 1 mL of sulfanilamide (0.33% in 20% glacial acetic acid) solution was added to 0.5 mL of the incubated mixture and allowed to stand for 5 min. Then, 1 mL of naphthyl ethylenediamine dihydrochloride (NED) (0.1% *w*/*v*) solution was added to the mixture and incubated for 30 min at 25 °C. The pink chromophore formed during the diazotization of nitrite ions with sulfanilamide and subsequent coupling with NED was measured using a spectrophotometer at λ = 546 nm against blank samples. NORS radical-scavenging activity was estimated for each biological replicate in triplicate (*n* = 9). The data were expressed as percentage inhibition by the following equation:NORS inhibition (%) = (A_control_ − A_test_)/(A_control_) × 100
where A_control_ means the absorbance of the control reaction and A_test_ represents the absorbance of a test reaction.

### 2.6. Untargeted Metabolomics Using UPLC-TOF-MS Analysis

For the untargeted metabolomics study, 200 μL of 80% methanol was added to the sample tube and vortexed for 30 s. Then, samples were kept at –40 °C for 1 h. After that, samples were vortexed for 30 s and centrifuged at 12,000 rpm for 15 min/4 °C. Finally, 150 μL of supernatant and 5 μL of DL-*o*-chlorophenylalanine (140 μg/mL) were transferred to the vial for LC-MS analysis. Separation was performed by an Ultimate 3000LC combined with Q Exactive MS (Thermo Fisher Scientific) and screened with ESI-MS. The LC system is composed of an ACQUITY UPLC HSS T3 (100 × 2.1 mm, 1.8 μm) with an Ultimate 3000LC. The mobile phase was composed of solvent A (0.05% formic acid–water) and solvent B (acetonitrile) with a gradient elution (0–1 min, 5% B; 1–12 min, 5–95% B; 12–13.5 min, 95% B; 13.5–13.6 min, 95–5% B; 13.6–16.0 min, 5% B). The flow rate of the mobile phase was 0.3 mL/min. The column temperature was maintained at 40 °C, and the sample manager temperature was set at 4 °C. Mass spectrometry parameters for ESI+ and ESI− mode are listed as follows:

ESI+: heater temp, 300 °C; sheath gas flow rate, 45 arb; aux gas flow rate, 15 arb; sweep gas flow rate, 1 arb; spray voltage, 3.0 kV; capillary temp, 350 °C; S-lens RF level, 30%.

ESI−: heater temp, 300 °C; sheath gas flow rate, 45 arb; aux gas flow rate, 15 arb; sweep gas flow rate, 1 arb; spray voltage, 3.2 kV; capillary temp, 350 °C; S-lens RF level, 60%.

### 2.7. Data Processing and Statistical Analysis

Raw data were acquired and aligned using the Compound Discover (3.0, Thermo Fisher Scientific) based on the *m*/*z* value and the retention time of the ion signals. Ions from ESI− or ESI+ were merged and imported into the SIMCA-P program (version 14.1). All multivariate and statistical analyses, including the metabolite set enrichment analysis (MSEA), principal components analysis (PCA), partial least-squares–discriminant analysis (PLS-DA), and heatmap of Pearson correlation analysis, were performed using MetaboAnalyst 5.0 online software (https://www.metaboanalyst.ca/MetaboAnalyst/home.xhtml). MSEA was carried out based on enrichment analysis by inputting the Human Metabolome Database (HMDB) ID to specify major chemical groups of identified metabolites. A data matrix with rows representing samples and columns describing features was prepared, and the resulting data were exported to an Excel (.csv) file (Microsoft Corporation, Redmond, WA, USA) for multivariate analysis. For statistical analysis, the data were transformed into a logarithmic base and Pareto scaling was performed. Principle components analysis (PCA) was first used as an unsupervised method to find whether there were real differences among muscadine genotypes throughout the developmental stages. Furthermore, supervised regression modeling was performed on the data set using PLS-DA to obtain the variable importance in the projection (VIP). The biomarkers were filtered and confirmed by combining the results of the VIP, |*p*|, and |*p*(corr)|, and the screened compounds were analyzed by the heatmap of Pearson correlation with antioxidant activities to find out the potential nutritional biomarkers.

The significant differences between experimental groups were determined based on analysis of variance (ANOVA) using SAS 9.1.3 software (Cary, NC, USA). Duncan’s multiple-range test analyzed the comparison of means, and differences were considered statistically significant when *p* < 0.05. The muscadine genotypes were compared to distinguish a possible difference in their modulation of antioxidant activities. The average antioxidant activity among different stages for each genotype was calculated and statistically tested for differences between genotypes. In another approach, the slope of the antioxidant activity curve for each genotype was calculated. Then, the slopes were statistically compared to disclose any significant differences at 95% confidence.

### 2.8. Metabolite Identification

The most significant metabolite MS/MS spectra were acquired and searched in the following databases: METLIN [29] (https://metlin.scripps.edu/landing_page.php?pgcontent=mainPage), the Human Metabolome Database [30] (www.hmdb.ca), Massbank [31] (https://massbank.eu/MassBank/Search), and MetFrag [32] (https://ipb-halle.github.io/MetFrag/). Additionally, based on necessity, further confirmation was acquired through comparisons with authentic standards, including retention times and MS/MS fragmentation patterns. Furthermore, the available raw data in the public database (MTBLS2877) describe that our study follows the guideline of metabolite identification parameters according to Fernie et al. [33]. The metabolic pathway map was constructed based on the relevant literature and the KEGG database (https://www.kegg.jp/kegg-bin/show_pathway?161345586717248/vvi01100.args).

## 3. Results and Discussion

### 3.1. Total Phenolic and Flavonoid Content

Metabolites play crucial roles in the physicochemical characteristics, quality parameters, and nutritional benefits of grapes [11]. The total phenolic content (TPC) and total flavonoid content (TFC) were assessed at different developmental stages of three muscadine genotypes, designated as LF, C5, and C6 (Figure 1A). The TPC level range was between 30.8 and 705.3 mg GAE/g FW amongst different muscadine genotypes and developmental stages. Relative to other stages, the RIP-S stage displayed the highest TPC levels (*p* > 0.05) without significant differences among genotypes. The levels of TPC during berry development of all genotypes were arranged in order from high to low abundance as follows: RIP-S > FS > pre-V ≅ V ≅ post-V > RIP-SF (Figure 1B). At the FS stage, the TPC level significantly differed amongst the three genotypes, with the C6 genotype exhibiting the highest TPC (393 mg GAE/g FW), followed by C5 and LF. At the pre-V stage, the LF genotype showed the highest TPC levels (230 mg GAE/g FW). TPC levels in C5 and C6 at the pre-V stage were considerably lower and were ~85% and ~79%, respectively, of those detected in LF. At the V stage, the C5 genotype recorded the highest TPC levels (177.3 mg GAE/g FW), which was noticeably higher than that detected in LF and C6 by 67% and 75%, respectively. At the post-V stage, the TPC level was considerably different amongst the three genotypes. The C5 genotype showed the highest TPC levels (117.2 mg GAE/g FW), while LF and C6 displayed 80% and 54% of the TPC detected in C5, respectively. Furthermore, the C5 genotype exhibited the highest TPC levels (51.7 mg GAE/g FW) in RIP-SF, which was extensively higher than that recorded in LF and C6 by 28% and 40%, respectively. Finally, the TPC levels were steadily abundant in the RIP-S stage, with an average content of ~679.9 mg GAE/g FW.

The TFC level range was between 2.6 and 71.4 mg QE/g FW among all muscadine samples. Relative to other stages, the FS stage showed the highest TFC levels (*p* > 0.05). Comparable to different genotypes, LF at the FS stage manifested the lowest TFC abundance (52.8 mg QE/g FW). The TFC levels in LF represented 74% and 79% of those detected in C6 and C5 genotypes, respectively. Based on TFC abundance, the FS stage was followed by the pre-V and RIP-S stages in the three genotypes (Figure 1C). No substantial differences were identified between V, post-V, and RIP-SF stages for the LF and C6 genotypes. However, C5 exhibited the following tendencies: V ≈ pre-V ≈ RIP-S > post-V ≈ RIP-SF. At the V stage, the TFC levels in the C5 genotype were considerably higher than those detected in C6 and LF by ~85%. Despite the low TFC levels at advanced post-V and RIP-SF developmental stages, the TFC in C5 genotype remained larger than in the other two genotypes. At the post-V stage, TFC levels in C5 were estimated at 4.7 mg QE/g DW, while TFC abundance in LF and C6 was 21% and 44%, respectively, less than in C5. In RIP-SF tissue, the TFC levels in C5 were estimated at 6.1 mg QE/g DW, whereas the TFC abundance in LF and C6 was 48% and 55%, respectively, less than in C5. Finally, in the RIP-S stage, TFC was relatively abundant, with the highest levels assessed in the LF genotype (29.6 mg QE/g DW). The C6 and C5 genotypes displayed only 74% and 61% of the TFC levels in LF, respectively. Despite the significant differences in TFC between LF and C5, a negligible variation was observed between LF and C6. Similar results of TFC levels among the three genotypes during ripening were previously reported [6].

Several studies have reported that the highest phenolic content in muscadine berries is mainly accumulated in seeds (60–70%), followed by skin (28–35%) and pulp (10%) [2,9,11]. However, there is still a lack of muscadine or bunch grape studies assessing TPC or TFC levels through berry development. Evaluation of numerous *V. vinifera* genotypes at the ripening stage indicated that the whole berry (including skin, flesh, and seeds) contains a wide range of total phenolic (95.3–686.5 mg GAE/100 g FW) and flavonoid (94.7–1055 mg QE/100 g FW) contents [34]. In our study, the average (sum of skin/flesh and seeds) values of TPC and TFC in ripe muscadine berries were estimated between 667.3 and 757.0 mg GAE/g FW and 24.3 and 32.8 mg QE/g FW, respectively. By comparing the two studies, it can be concluded that TPC and TFC levels in muscadine grapes are remarkably higher than those detected in *V. vinifera* grapes by ~182- and ~5-fold, respectively. The results obtained in our study suggested that the muscadine grape could be used as a promising source for the development of any functional food or bioactive products due to its higher phenolic metabolite content.

### 3.2. Total Antioxidant Activities

The antioxidant activities of natural crude extracts from plant/food sources are multifunctional and dependent on the structure–activity relationship of the bioactive metabolite compositions. Therefore, defining the antioxidant potentials using different antioxidant methods would provide various aspects of their antioxidant capacities [35]. In this study, the antioxidant activities of muscadine grapes were determined using five different antioxidant assays (Figure 2).

The main muscadine developmental stages that contributed to DPPH radical-scavenging activity were RIP-S (40%), FS (28%), and pre-V (15%). However, V (7%), post-V (6%), and RIP-SF (4%) stages exhibited relatively less percentages (Figure 2A). Similarly, analysis of FRAP radical-scavenging activity revealed that the highest antioxidant activity is associated with RIP-S (37%), FS (32%), and pre-V (17%) stages. Nevertheless, the V (6%), post-V (5%), and RIP-SF (3%) stages were less involved (Figure 2B). The order of muscadine developmental stages was reversed in ABTS, CUPRAC, and NORS antioxidant activities. The intermediate muscadine developmental stages of pre-V (21%), post-V (21%), and V (20%), along with RIP-S (19%), were the main contributors to ABTS activity. Nonetheless, RIP-SF (15%) and FS (4%) stages were found less important (Figure 2C). CUPRAC was more incorporated with the advanced developmental stages of post-V (38%), RIP-S (30%), and RIP-SF (21%); however, early immature stages of V (5%), pre-V (4%), and FS (2%) had a negligible influence (Figure 2D). Likewise, NORS activity was connected with advanced muscadine developmental stages of post-V (26%), RIP-SF (25%), and RIP-S (19%) rather than other earlier developmental stages of V (13%), pre-V (13%), and FS (3%) (Figure 2E). The previous analysis suggested a potential overlapped originator between DPPH and FRAP antioxidant activities and between CUPRAC and NORS activities.

The muscadine genotypes were compared to identify a potential difference in their modulation of antioxidant activities. Statistical analysis of the average and slope profile of the different antioxidant activities suggested insignificant differences between the three genotypes investigated in this study for FRAP, CUPRAC, and NORS. However, remarkable differences between genotypes were detected in the case of DPPH and ABTS activities. Despite the slight differences in the average DPPH activity between the LF and C6 genotypes, pronounced differences were identified between C5/LF and C5/C6. The analysis demonstrated a significantly higher DPPH activity average in the C5 genotype, estimated at 42.0%. However, the average DPPH activity in LF and C6 was dramatically lower than C5 by 35% and 47.9%, respectively. The slope of the DPPH activity curve in the LF, C5, and C6 genotypes was estimated at –0.5, –0.24, and –0.63, respectively. Statistical analysis revealed a slight difference between LF/C5 and LF/C6 slopes. However, a definite difference between C5/C6 slopes was identified, suggesting a distinct functional model between these two genotypes.

Similarly, the average ABTS activity was estimated at 77.6%, 82.7%, and 68.8% for LF, C5, and C6, respectively. No visible differences were noted between LF/C5 and LF/C6 means. However, a considerable variation was detected between C5/C6 means, suggesting that the C5 genotype generally manifests higher ABTS activity than C6. It is tempted to highlight that no significant differences were identified in the ABTS activity curve slope between the three genotypes, suggesting a similar activity profile.

### 3.3. The Metabolite Profiling of Muscadine Grape Genotypes at Selected Developmental Stages

The untargeted metabolomics profile of muscadine genotypes was performed for selected developmental stages of FS, V, RIP-SF, and RIP-S due to the significant variability in their total metabolite contents compared with the other stages of pre-V and post-V. A total of 329 compounds were identified in all samples. They were distributed as follows: 152 compounds were identified in ESI+ mode, 70 compounds in ESI− mode, and 107 compounds in ESI+ and ESI− modes (Appendix A). Metabolite set enrichment analysis (MSEA) was conducted to classify the chemical groups of all identified compounds (Figure 3). The bar chart represents the chemical classifications of the identified metabolite sets (top 25) under the chemical structure metabolite set library category (Figure 3A). As shown in the bar chart, among the top 25 chemical classes, the primary chemical groups with a higher *p*-value were mainly amino acids, trichloroacetic acids, dicarboxylic acids, hydroxycinnamic acids, primary amides, oxo fatty acids, pyridinecarboxylic acids, phenylacetic acids, and flavonoid glycosides. The colors in the interactive pie chart designate each chemical group relative to the total number of compounds (Figure 3B). Among the 15 chemical groups, the highest number of compounds were mainly described by amino acids (red), followed by dicarboxylic acids (purple), flavones (blue), organic dicarboxylic acids (clover green), TCA acids (lizard green), and primary amides (orange).

Other reports have indicated that fruit acids, carbohydrates, and phenolics are the most abundant chemical constituents in grapes [36].

The metabolite data of this study were further subjected to multivariate analysis using the principal components analysis (PCA) approach to obtain an overview of the differences in the metabolite among the three genotypes at different developmental stages (Figure 4). As shown in the PCA 2D score plot (Figure 4A), the first and second principal components explained 70.2% and 20.6% of the variation, respectively. All muscadine samples were clustered into separate individual groups of FS and RIP-S. Conversely, a close aggregation was visualized between V and RIP-SF, suggesting the inter-developmental stage variations in the metabolite profiles of muscadine grape genotypes. The PCA scatter plot suggested that catechin, epicatechin-3-gallate, gallic acid were the primary muscadine metabolites contributing to the RIP-S stage (Figure 4B). However, malonic acid and tartaric acid were the central donors to the FS stage. Finally, malic acid was the predominant contributor in the V and RIP-SF stages. Throughout muscadine fruit developmental phases, grape berries undergo dramatic changes in physiological and biochemical attributes [17]. Starting at the fruit-set (FS) stage, a significant accumulation of organic acids (i.e., tartaric acid, malic acid, and malonic acid) occurs and proceeds until the veraison (V) stage, as the grape berry remains immature [17].

The reduction of organic acid and the accumulation of sugars at the ripening stage coincide with enhanced catechins in seeds, notably catechin, epicatechin, procyanidins, and their polymers [9,17].

### 3.4. Multivariate Analysis of Candidate Metabolites and Antioxidant Activities of Muscadine Grapes

PLS-DA of muscadine berry metabolites was performed to obtain the candidate compounds based on VIP scores. As a result, 27 candidate biomarker compounds (VIP > 0.5) were determined among all identified metabolites in muscadine berries (Table 1). As the untargeted metabolomics identified many metabolites (total 329), the VIP score value greater than 0.5 was used to represent the significant contributing compound of muscadine grapes. The biomarker compounds in muscadine berries were dominated by malic acid, malonic acid, tartaric acid, oxalic acid, trehalose, catechin, gallic acid, L-arginine, citric acid, oxalacetic acid, quinic acid, 4-acetamidobutanoic acid, glutamine, dihydrouracil, 2-furanmethanol, pyroglutamic acid, serine, ornithine, proline, glycerophosphocholine, epicatechin-3-gallate, L-leucine, fumaric acid, 7-methylxanthine, ellagic acid, hydroxypropionic acid, and tiglic acid. These compounds can be collectively categorized into organic acids, flavonoids, phenolic acid, and amino acids.

It has been reported that muscadine grape phenolics mainly consist of phenolic acid derivatives, flavonoids, gallic acid, ellagic acid derivatives, and tannins [10,11]. Nevertheless, there are still limited data available to compare the untargeted metabolomics of muscadine grape genotypes at different developmental stages.

Using the untargeted metabolome strategy and assessing the antioxidant activity of the same berry context/genotype at different developmental stages can facilitate identifying a particular metabolome associated with specific antioxidant activity. Consequently, the 27 candidate compounds and the five different types of antioxidant activities of muscadine berries were plotted with a heatmap. As shown in Figure 5, higher antioxidant activities were detected in the RIP-S stage, followed by RIP-SF, FS, and V stages of muscadine genotypes. All the five antioxidant activities were higher at the RIP-S stage of all genotypes, whereas CUPRAC and NORS were higher at RIP-SF, and ABTS was higher in C5 and LF than C6. The FS stage of all genotypes exerted higher DPPH and FRAP activities, whereas at the V stage, all genotypes exhibited ABTS activity but DPPH activity was only found in C5.

Moreover, at the RIP-S stage of all genotypes, oxalacetic acid, epicatechin-3-gallate, catechin, citric acid, gallic acid, 4-acetamidobutanoic acid, L-leucine, and tiglic acid were more abundant. Ellagic acid, catechin, ornithine, proline, and 2-furanmethanol were dominant in C5, whereas trehalose was dominant in LF and C6 genotypes. At the RIP-SF stage of all genotypes, 7-methylxanthine, oxalacetic acid, trehalose, quinic acid, citric acid, ornithine, proline, 2-furanmethanol, and hydroxypropionic acid were the major compounds. L-arginine, glutamine, pyroglutamic acid, dihydrouracil, and serine were dominant in LF and C6 genotypes. Malic acid and tiglic acid were predominant in LF and C6 genotypes, whereas ellagic acid, malic acid, and fumaric acid were abundant in the C5 genotype. At the FS stage of all genotypes, the significant metabolites were tartaric acid, malonic acid, oxalic acid, glycerophosphocholine, gallic acid, and tiglic acid. While ellagic acid and pyroglutamic acid were abundant in C5/C6 genotypes, 4-acetamidobutanoic acid, glutamine, and serine were higher in C5, and 4-acetamidobutanoic acid was more elevated in LF. At the V stage, malic acid and fumaric acid were higher in all genotypes. In contrast, 7-methylxanthine, oxalacetic acid, citric acid, glutamine, ornithine, pyroglutamic acid, proline, dihydrouracil, serine, 2-furanmethanol, and hydroxypropionic acid were higher in LF and C6 genotypes. Tartaric acid, oxalic acid, and malonic acid were abundant in C5 and C6 genotypes. Abundances of trehalose, quinic acid, and L-arginine were only found in LF, while ellagic acid and tiglic acid were highly detected in the C5 genotype.

Additionally, a heatmap Pearson’s correlation analysis was performed to determine the correlation between candidate compounds and antioxidant activities (Figure 6). The yellow, green, and blue colors in the plot represent the higher-to-lower correlation intensity. According to the Pearson correlation coefficient, cluster analysis suggested that DPPH and FRAP antioxidant activities had a similar pattern compared to other types of antioxidant activities. This evaluation also complies with our antioxidant results, as presented in Figure 2. A detailed view of Pearson’s correlation coefficient among candidate compounds with individual antioxidant activity is separately shown in Figure 7A. It was observed that gallic acid and 4-acetamidobutanoic acid were highly correlated with DPPH and FARP antioxidant activities. In addition, epicatechin-3-gallate, catechin, and L-leucine were positively correlated with only DPPH. In comparison, oxalacetic acid and citric acid were regarded as highly associated with ABTS radical-scavenging activity. Epicatechin-3-gallate, catechin, and L-leucine also showed a higher positive correlation with CUPRAC antioxidant potential. However, oxalacetic acid, trehalose, quinic acid, and citric acid were highly correlated with NORS activity. In the meantime, 7-methylxanthine and hydroxypropionic acid showed a positive correlation with NORS. Among these nutritional biomarkers, nine compounds were positively associated with different antioxidant activities: gallic acid, citric acid, quinic acid, catechin, trehalose, epicatechin-3-gallate, oxalacetic acid, 4-acetamidobutanoic acid, and L-leucine.

The intensity of absorbance (MAU.s) for these nine nutritional biomarkers associated with the particular antioxidant capacity of muscadine grape genotypes at different developmental stages is illustrated in Figure 7B. The results suggested that total intensity was higher in the RIP-S stage, followed by RIP-SF, V, and FS stages. For the RIP stage, catechin and trehalose were dominant in RIP-S and RIP-SF stages, respectively, in a genotype-independent manner. For the V stage, trehalose, oxalacetic acid, and citric acid were predominant in the LF and C6 genotypes. However, only oxalacetic acid and citric acid were dominating in the C5 genotype. For the FS stage, 4-acetamidobutanoic acid and gallic acid were highly represented in all muscadine genotypes.

The nine nutritional biomarkers found in this study are well-known bioactive metabolites, naturally found in plant/food-based sources and are associated with numerous health benefits. Among these nutritional biomarkers, gallic acid is a plant-based phenolic acid derivative, consisting of an aromatic ring, three phenolic hydroxyl groups, and a carboxylic acid group [37,38]. The antioxidant potentials of gallic acid have been reported by several other studies [37,38,39,40]. Similar to our findings, another study reported the capacity of gallic acid for FRAP activity. In that study, gallic acid showed higher activity than several other standard antioxidants, including ascorbic acid, Trolox, and uric acid [40]. The antioxidant principle of gallic acid probably involves the hydrogen-donating mechanism [39]. It could also be due to the arrangement of three hydroxyl groups, which are bonded to the aromatic ring in an ortho-position concerning each other. Another antioxidant potential was described for 4-acetamidobutanoic acid, the primary metabolite of GABA, a naturally occurring non-protein amino acid. Along with several other health benefits, the antioxidant potential of GABA derivatives by DPPH [41,42,43] and FRAP assay [44] have also been reported. Another identified nutritional biomarker in our study is citric acid, a natural antioxidant agent found in many fruits. As a potent scavenger, citric acid prevents cellular oxidation by scavenging ROS such as hydrogen peroxide, hydroxyl, alkoxy, peroxide radicals, and superoxide anions. It was demonstrated that citric acid stabilizes ROS through the direct transfer of hydrogen atoms from the antioxidant molecule to prevent ROS-mediated cell damage [45]. A potential antioxidant compound identified in this study is catechins, one of the most naturally occurring potent compounds for quenching ROS. The antioxidant mechanism underlying catechins involves the donation of the phenolic OH- group to reduce free radicals [46]. One more antioxidant determinant is quinic acid, which has been reported as a potential antioxidant stimulator associated with SAR [47,48,49]. Trehalose was also identified as the nutritional biomarker in this study. It is a non-reducing disaccharide naturally present in many plant species. It has been recently reported that trehalose protects cellular damage against oxidative stress [50]. Another biomarker antioxidant is L-leucine, a branched-chain amino acid that has also been reported to improve antioxidant activities [51].

## 4. Conclusions

The metabolome profiling and antioxidant activities of muscadine grape genotypes at different developmental stages were investigated based on untargeted metabolomics combined with multivariate analysis. Untargeted metabolomics identified 329 metabolites of several functional chemical groups, including organic acids, amino acids, flavonoids, sugar, and others in muscadine grapes. Statistical analysis suggested that variations among berry developmental stages are more significant than those existing between genotypes. Correlation analysis indicated that each antioxidant capacity of muscadine grapes is positively correlated with a particular set of nutritional biomarkers. The nutritional biomarkers with higher antioxidant activities were mainly determined as gallic acid, 4-acetamidobutanoic acid, citric acid, quinic acid, and trehalose. Moreover, the relative contents of all nutritional biomarkers in muscadine genotypes were higher in ripening seeds. Our finding suggests that the muscadine grape contains several nutritional metabolites that could be of great research interest for food/pharmaceutical or nutraceutical sectors. However, there are limitations to the metabolomics approach to account for entire metabolite profiles and colorimetric antioxidant assays to determine the direct physiological function. In this regard, we propose that further detailed studies involving the determination of the maximum range/content of muscadine metabolites and pharmacological efficiencies be performed. Despite this, the present study provided a framework for fundamental biochemical analyses and an informed database that can also be used for future characterization of larger muscadine grape populations toward exploring genetic markers and QTLs associated with certain bioactive compounds. Indeed, our results suggested that two of the five antioxidant assays (e.g., DPPH and ABTS) are able to distinguish the investigated genotypes. Accordingly, only these assays will be used for further phenotypic–genotypic association studies. This knowledge is of great value for plant breeders aiming at developing new muscadine grape varieties with improved nutraceutical values.

## Figures and Tables

**Figure 1 antioxidants-10-00914-f001:**
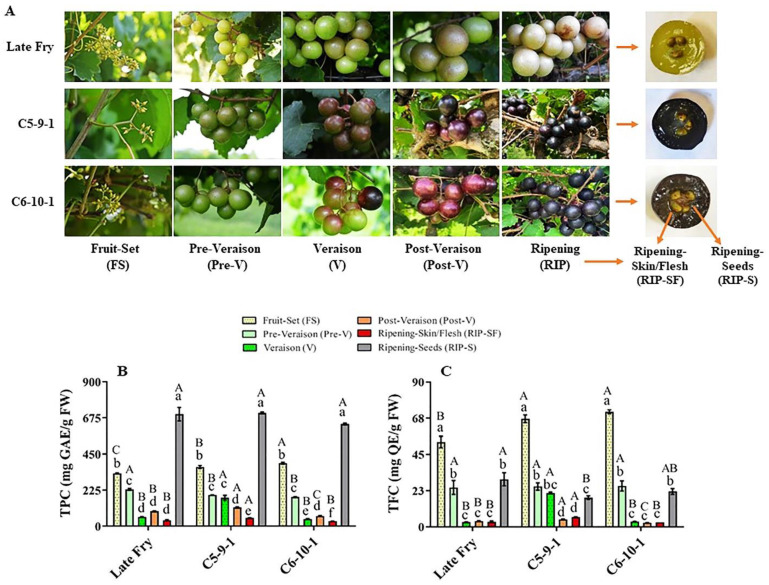
(**A**) A representative image of muscadine grape genotypes Late Fry, C5-9-1, and C6-10-1 during developmental stages. The developmental stages include fruit-set (FS), pre-veraison (pre-V), veraison (V), post-veraison (post-V), and ripening (RIP). The ripening stage was separated into skin/flesh (RIP-SF) and seeds (RIP-S). Accumulation of (**B**) total phenolic content (TPC) and (**C**) total flavonoid content (TFC) of muscadine grape genotypes at different developmental stages. Colors representing each developmental stage are indicated. The experiments were carried out in three biological replicates, and each replicate was repeated three times (*n* = 9). Data represent the mean values ± SD (*n* = 3). The different lowercase letters represent the significant differences among developmental stages of an individual genotype, and the different uppercase letters refer to the significant differences between genotypes of a particular developmental stage, according to Duncan’s multiple-range test (*p* > 0.05).

**Figure 2 antioxidants-10-00914-f002:**
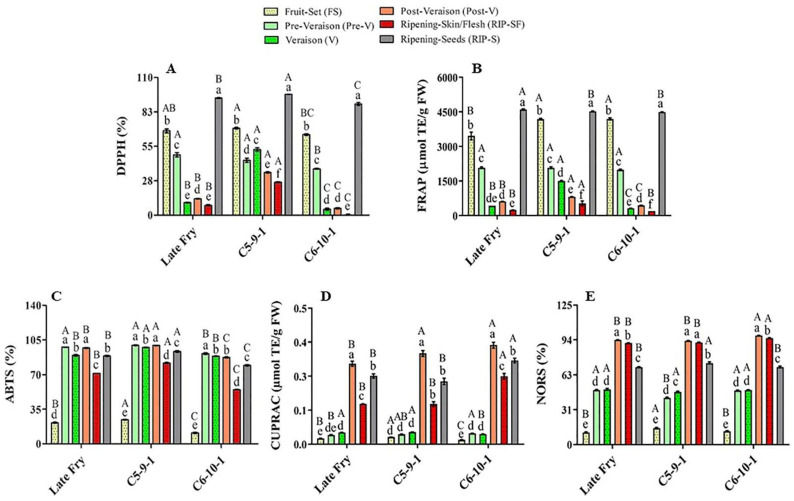
Antioxidant activities of muscadine grape genotypes at different developmental stages were determined using DPPH (**A**), FRAP (**B**), ABTS (**C**), CUPRAC (**D**), and NORS (**E**) assays. The experiments were carried out in three biological replicates, and each replicate was repeated three times (*n* = 9). Data represent the mean values ± SD (*n* = 3). The different lowercase letters represent the significant differences among developmental stages of an individual genotype, and the different uppercase letters refer to the significant differences between genotypes of a particular developmental stage, according to Duncan’s multiple-range test (*p* > 0.05).

**Figure 3 antioxidants-10-00914-f003:**
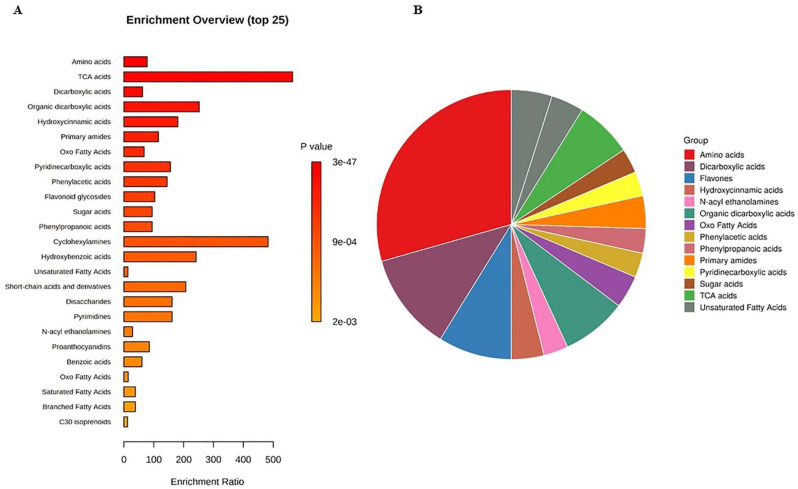
(**A**) Bar chart and (**B**) interactive pie chart of the chemical classification of muscadine metabolites at selected developmental stages using metabolite set enrichment analysis (MSEA). Colors in the bar plot describe the *p*-value. The red and orange colors signify the high and low values, respectively. The lines indicate the enrichment ratio, which was computed by hits/expected, where hits = observed hits and expected = expected hits. The colors in the interactive pie chart designate each chemical group relative to the total number of compounds.

**Figure 4 antioxidants-10-00914-f004:**
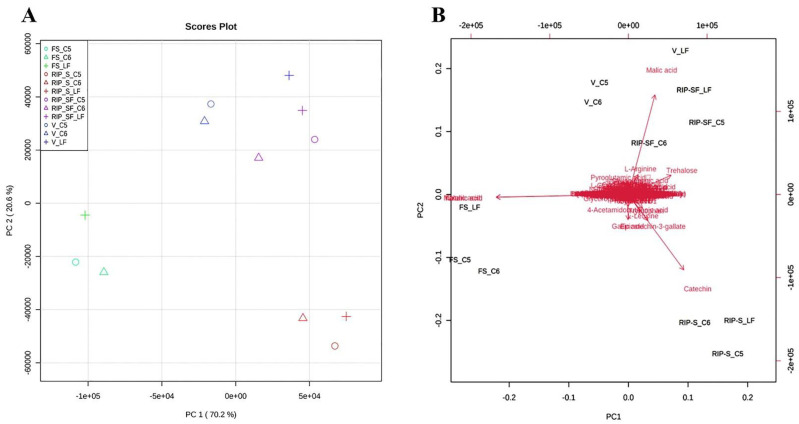
Principal components analysis (PCA) 2D score plot (**A**) and biplot (**B**) of the muscadine metabolites at selected developmental stages. The different short abbreviations in the biplot manifest the scores of the observations (i.e., muscadine genotypes). The vectors that point toward the same direction correspond to the variables (i.e., metabolites) with similar response profiles. The green color of the oval, triangle, and cross shape represents FS_C5, FS_C6, and FS_LF; blue represents V_C5, V_C6, and V_LF; purple represents RIP-SF_C5, RIP-SF_C6, and RIP-SF_LF; and red represents RIP-S_C5, RIP-S_C6, and RIP-S_LF. FS_LF: fruit-set_Late Fry; FS_C5: fruit-set_C5-9-1; FS_C6: fruit-set_C6-10-1; V_LF: veraison_Late Fry; V_C5: veraison_C5-9-1; V_C6: veraison_C6-10-1; RIP-SF_LF: ripening-skin/flesh_Late Fry; RIP-SF_C5: ripening-skin/flesh_C5-9-1; RIP-SF_C6: ripening-skin/flesh_C6-10-1; RIP-S_LF: ripening-seeds_Late Fry; RIP-S_C5: ripening-seeds_C5-9-1; RIP-S_C6: ripening-seeds_C6-10-1.

**Figure 5 antioxidants-10-00914-f005:**
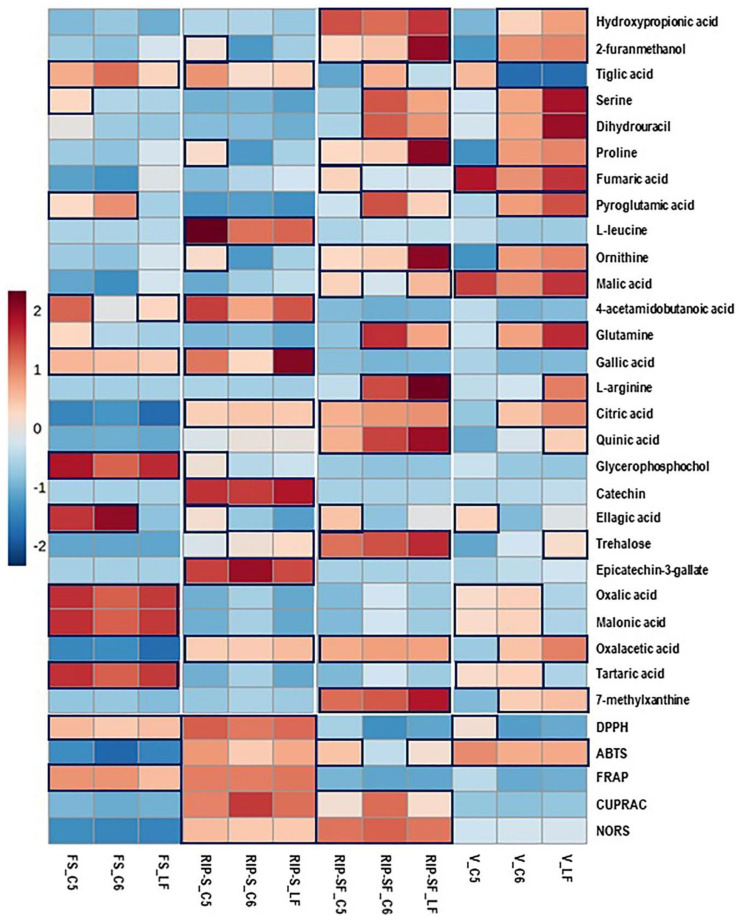
Heatmap analysis of candidate metabolites (VIP > 0.5) obtained by partial least-squares–discriminant analysis (PLS-DA) and antioxidant activities of muscadine genotypes at selected developmental stages. Each column refers to the muscadine genotype at different developmental stages, and each row indicates the metabolites and antioxidant activities. The red and blue colors in the plot describe high and low intensities, and the values range from –2 to +2. The higher the red color intensity (from +1 to +2 values), the higher the metabolite contents and antioxidant activities; in contrast, the higher blue color intensity (from –1 to –2 values) represents lower metabolite contents and antioxidant activities. FS_LF: fruit-set_Late Fry; FS_C5: fruit-set_C5-9-1; FS_C6: fruit-set_C6-10-1; V_LF: veraison_Late Fry; V_C5: veraison_C5-9-1; V_C6: veraison_C6-10-1; RIP-SF_LF: ripening-skin/flesh_Late Fry; RIP-SF_C5: ripening-skin/flesh_C5-9-1; RIP-SF_C6: ripening-skin/flesh_C6-10-1; RIP-S_LF: ripening-seeds_Late Fry; RIP-S_C5: ripening-seeds_C5-9-1; RIP-S_C6: ripening-seeds_C6-10-1.

**Figure 6 antioxidants-10-00914-f006:**
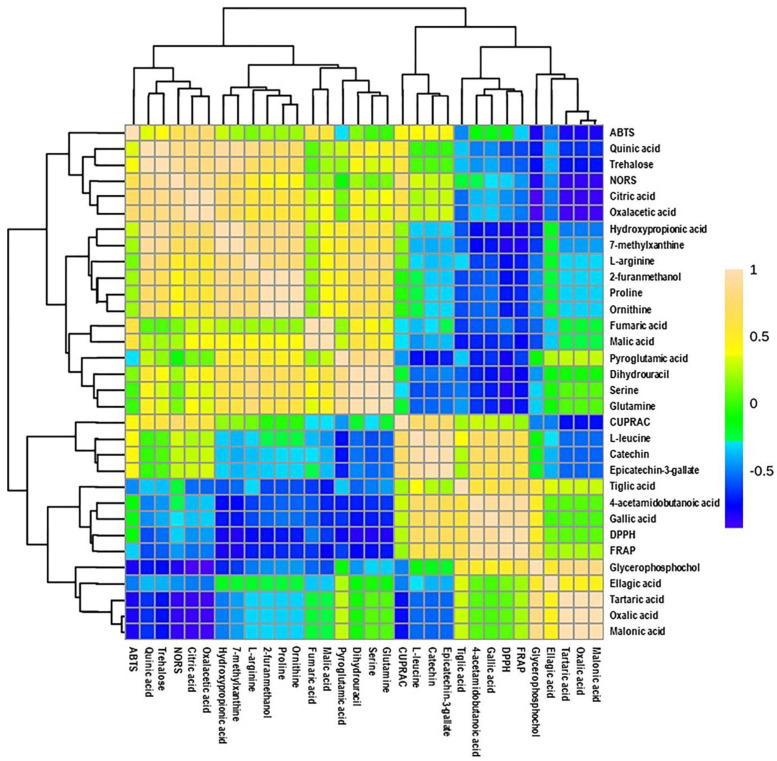
Heatmap of Pearson correlation between candidate metabolites (VIP > 0.5) with antioxidant activities of muscadine genotypes at selected developmental stages. Correlation values range from –1 to +1. The values close to +1 represent the higher positive correlation, whereas values closer to zero mean there is no linear trend between the variables; values close to –1 represent the negative correlation between variables. FS_LF: fruit-set_Late Fry; FS_C5: fruit-set_C5-9-1; FS_C6: fruit-set_C6-10-1; V_LF: veraison_Late Fry; V_C5: veraison_C5-9-1; V_C6: veraison_C6-10-1; RIP-SF_LF: ripening-skin/flesh_Late Fry; RIP-SF_C5: ripening-skin/flesh_C5-9-1; RIP-SF_C6: ripening-skin/flesh_C6-10-1; RIP-S_LF: ripening-seeds_Late Fry; RIP-S_C5: ripening-seeds_C5-9-1; RIP-S_C6: ripening-seeds_C6-10-1.

**Figure 7 antioxidants-10-00914-f007:**
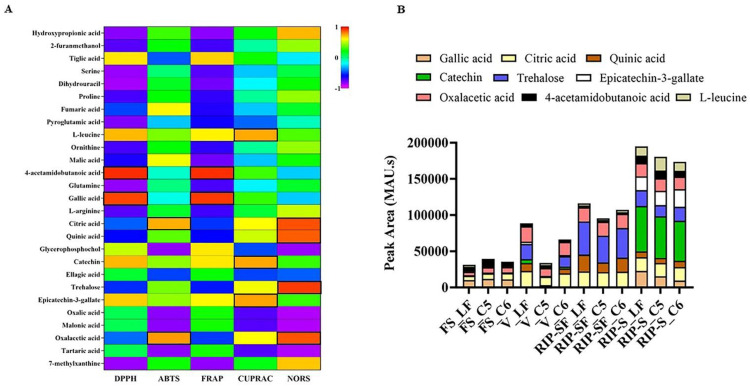
Pearson correlations (**A**,**B**) the intensity of absorbance (MAU.s) values of nutritional biomarkers of muscadine genotypes at selected developmental stages. FS_LF: fruit-set_Late Fry; FS_C5: fruit-set_C5-9-1; FS_C6: fruit-set_C6-10-1; V_LF: veraison_Late Fry; V_C5: veraison_C5-9-1; V_C6: veraison_C6-10-1; RIP-SF_LF: ripening-skin/flesh_Late Fry; RIP-SF_C5: ripening-skin/flesh_C5-9-1; RIP-SF_C6: ripening-skin/flesh_C6-10-1; RIP-S_LF: ripening-seeds_Late Fry; RIP-S_C5: ripening-seeds_C5-9-1; RIP-S_C6: ripening-seeds_C6-10-1.

**Table 1 antioxidants-10-00914-t001:** Candidate metabolites (VIP > 0.5) of muscadine grape genotypes at selected developmental stages.

No.	Compounds	VIP	Chemical Class	FS	V	RIP-SF	RIP-S
1	Malic acid	11.16	Dicarboxylic acid	YES	YES	YES	YES
2	Malonic acid	7.35	Dicarboxylic acid	YES	YES	YES	YES
3	Tartaric acid	7.34	Carboxylic acid	YES	YES	YES	YES
4	Oxalic acid	7.33	Dicarboxylic acid	YES	YES	YES	YES
5	Trehalose	3.04	Disaccharide	YES	YES	YES	YES
6	Catechin	2.27	Flavonoid	YES	YES	YES	YES
7	Gallic acid	2.06	Phenolic acid	YES	YES	YES	YES
8	L-arginine	1.97	Essential amino acid	YES	YES	YES	YES
9	Citric acid	1.65	Tricarboxylic acid	YES	YES	YES	YES
10	Oxalacetic acid	1.63	Oxodicarboxylic acid	YES	YES	YES	YES
11	Quinic acid	1.53	Cyclohexanecarboxylic acid	YES	YES	YES	YES
12	4-Acetamidobutanoic acid	1.04	Gamma amino acids	YES	YES	YES	YES
13	Glutamine	0.98	Amino acid	YES	YES	YES	YES
14	Dihydrouracil	0.95	Pyrimidones	YES	YES	YES	YES
15	2-Furanmethanol	0.86	Heteroaromatic compound	YES	YES	YES	YES
16	Pyroglutamic acid	0.82	Alpha amino acids	YES	YES	YES	YES
17	Serine	0.81	Amino acid	YES	YES	YES	YES
18	Ornithine	0.79	Amino acid	YES	YES	YES	YES
19	Proline	0.79	Amino acid	YES	YES	YES	YES
20	Glycerophosphocholine	0.75	Glycerophosphocholine	YES	YES	YES	YES
21	Epicatechin-3-gallate	0.71	Flavan-3-ol	YES	YES	YES	YES
22	L-leucine	0.67	Essential amino acid	YES	YES	YES	YES
23	Fumaric acid	0.61	Organic compound	YES	YES	YES	YES
24	7-Methylxanthine	0.55	Purines	YES	YES	YES	YES
25	Ellagic acid	0.53	Hydrolyzable tannin	YES	YES	YES	YES
26	Hydroxypropionic acid	0.52	Alpha hydroxy acid	YES	YES	YES	YES
27	Tiglic acid	0.51	Monocarboxylic unsaturated organic acid	YES	YES	YES	YES

FS: fruit-set stage; V: veraison stage; RIP-SF: ripening skin/flesh stage; RIP-S: ripening seeds stage.

## Data Availability

The data presented in this study are available to anyone on request.

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
