# Peer review of "Untargeted Metabolomics and Antioxidant Capacities of Muscadine Grape Genotypes during Berry Development"

_antioxidants, 2021, doi:10.3390/antiox10060914_

Round 1
Reviewer 1 Report
In the present manuscript, authors performed an untargeted metabolomics and an evaluation of antioxidant capacities of three different muscadine grape genotypes during their development stage. The results are interesting and provide additional information on the nutritional composition, in terms of bioactives, of this berry.
Specific comments are reported below:
-Lines 37-49: this part of introduction should be removed since does not add any relevant information within the context of the paper.
-Lines 64-66: I agree with the authors; however, this statement requires a reference.
-Lines 530-532 and along the manuscript: I suggest to replace “antioxidant biomarkers” with other terminology e.g. “nutritional biomarkers”. The term “antioxidants” refers to a specific biological/functional activity that it has not been documented/evaluated for these components in the present manuscript. In fact, the assays used to evaluate antioxidant activity are colorimetric and indirect, and they do not refer to a physiological function.
-Lines 565-566: This sentence must be removed or rephrased; this paper is not able to provide data about the potential functional properties of muscardine grapes.
Author Response
In the present manuscript, the authors performed untargeted metabolomics and an evaluation of antioxidant capacities of three different muscadine grape genotypes during their development stage. The results are interesting and provide additional information on the nutritional composition, in terms of bioactivities, of this berry.
We thank the Reviewer for the time invested and the numerous comments that considerably helped us to improve our manuscript. We appreciate the Reviewer’s overall feedback on the importance of the scientific objective addressed in the present study.
Specific comments are reported below:
- Lines 37-49: this part of the introduction should be removed since does not add any relevant information within the context of the paper.
This part is removed as per the reviewer's suggestion.
- Lines 64-66: I agree with the authors; however, this statement requires a reference.
The reference is added as per the reviewer's suggestion (ref. number 15).
- Lines 530-532 and along with the manuscript: I suggest replacing “antioxidant biomarkers” with other terminology e.g. “nutritional biomarkers”. The term “antioxidants” refers to a specific biological/functional activity that has not been documented/evaluated for these components in the present manuscript. In fact, the assays used to evaluate antioxidant activity are colorimetric and indirect, and they do not refer to a physiological function.
The “antioxidant biomarkers” is changed to “nutritional biomarkers” throughout the manuscript as per the reviewer's suggestion.
- Lines 565-566: This sentence must be removed or rephrased; this paper is not able to provide data about the potential functional properties of muscadine grapes.
The sentence is removed as per the reviewer's suggestion.
Reviewer 2 Report
The manuscript by Darwish et al. reports metabolite profiles and antioxidant activities of muscadine grape plants at different berry developmental stages. Using untargeted metabolomics approach based on LC-MS, the authors detected several hundreds of metabolites in this species, including amino and organic acids, sugars, and phenolics. Subsequent multivariate analysis identified 27 biomarker metabolites, and correlation analysis between the biomarkers and antioxidant activities was performed. The results indicated that the antioxidant capacity was associated with a particular set of metabolites. However, in my opinion, the details of experimental setup and metabolomic analyses are lacking. My major comments are as follows:
Aim of this study is not clear for me
Why did you select the three muscadine grape genotypes? What are these genotypes representatives for? Why did you investigate the metabolite profiles and antioxidant activities at different developmental stages?
Abstract
It might be a good idea to add information about your instrument to perform metabolite profiling. In addition, I think that VIP > 0.5 is too arbitrary. Could you show the reasonable explanation, please?
Methods
This journal says “The full experimental details must be provided so that the results can be reproduced.” You should carefully describe all the methods.
(1) There was no descriptions about the plant growth condition.
(2) There was no information about the metabolite identification and annotation in untargeted metabolomics using UPLC-TOF-MS analysis.
(3) Please describe the parameters for metabolite annotation according to the guideline by Fernie et al. 2011 (Plant Cell 23, 2477-2482). We cannot evaluate the correlation between metabolites and antioxidant activities in this study.
(4) L109 – TPC and TFC, the authors should explain each of your abbreviations the first time it appears in the main text.
(5) Statistical data analysis - You should describe PCA/PLS-DA in details, including data transform and scaling. Regarding MetaboAnalyst, the version and the parameters used are also important to reproduce the data analysis.
Figures and Tables
Fig. 4 – The resolution of this figure is too low. The font size is also small. The authors should describe all the colors/symbols (circles, triangles, and plus).
Fig. 5 – The authors should describe red and blue color values. For example, what does the value -2 mean? You mentioned “Data represent mean values ± SD (n = 3)” in this figure legend (L502-503). What does it mean?
Conclusions
This section is too short. Basically, the authors have considered only UPLC-MS based untargeted metabolites for the study which does not account for all metabolite profile of this species. How do we use the results of the present study for further QTL analysis and plant breeding?
Data availability
No (raw) data are available. You should make your data available in a public database, such as Metabolomics Workbench (http://www.metabolomicsworkbench.org/) and MetaboLights (http://www.ebi.ac.uk/metabolights/).
Author Response
The manuscript by Darwish et al. reports metabolite profiles and antioxidant activities of muscadine grape plants at different berry developmental stages. Using untargeted metabolomics approach based on LC-MS, the authors detected several hundreds of metabolites in this species, including amino and organic acids, sugars, and phenolics. Subsequent multivariate analysis identified 27 biomarker metabolites, and correlation analysis between the biomarkers and antioxidant activities was performed. The results indicated that the antioxidant capacity was associated with a particular set of metabolites. However, in my opinion, the details of the experimental setup and metabolomic analyses are lacking. My major comments are as follows:
We thank the Reviewer for the time invested and the numerous comments that helped to improve our manuscript. We appreciate the Reviewer’s overall feedback on the importance of the scientific objective addressed in the present study.
The aim of this study is not clear for me.
Why did you select the three muscadine grape genotypes? What are these genotypes representatives for? Why did you investigate the metabolite profiles and antioxidant activities at different developmental stages?
- The aim of this study was to investigate the untargeted metabolomics and antioxidant potentials of three muscadine genotypes throughout six developmental stages to provide a comprehensive database of the muscadine metabolite’s types, relative content, and nutritional benefits.
- Three muscadine grape genotypes, including one standard cultivar, ‘Late Fry (LF)’ as well as two breeding lines, ‘C5-9-1 (C5)’ and ‘C6-10-1(C6)’ were used in this study. In the materials and methods section (lines 109 -111 based on originally submitted version), the author’s mentioned that “The three genotypes were selected according to their diversity in their total phenolic content and total flavonoid content, as well as DPPH radical scavenging activity at ripening stage [6]” (Lines 100-102 in the revised manuscript). The authors also added the corresponding reference of their previous study.
- As the aim of this study was to determine the metabolite profiling and antioxidant activities of muscadine genotypes at different developmental stages. The authors investigated the changes of metabolite profiles and their correlation with antioxidant activities of muscadine grapes at different major developmental stages. As we know, colorimetric antioxidant activities are used to evaluate the screening of potential nutritional biomarkers from any food-/plant-based sources. Therefore, each genotype at each developmental stage was evaluated for untargeted metabolomics and five different antioxidant activities to find out the changes in potential nutritional biomarkers and positive correlation. There is still limited information regarding the metabolomics study along with antioxidant properties of muscadine genotypes during the whole developmental stages. Thus, this study provides a valuable database to target biomarkers compounds of muscadine genotypes during developmental stages.
Abstract
It might be a good idea to add information about your instrument to perform metabolite profiling. In addition, I think that VIP > 0.5 is too arbitrary. Could you show a reasonable explanation, please?
- The instrument name is added in the revised manuscript as per the reviewer's suggestion (Lines 17-19).
- As the untargeted metabolomics identified many metabolites (total 329), the VIP score value greater than 0.5 was used to represent the significant contributing compound of muscadine grapes (Lines 441-444).
- Generally, the VIP > 1 is used for selecting relevant variables; according to the PLS-DA model, constant cut-off threshold values of VIP are not sometimes suitable for every data structure, a new cut-off threshold for VIP in classification task can be used to present data (Akarachantachote et al., 2013).
Methods
This journal says “The full experimental details must be provided so that the results can be reproduced.” You should carefully describe all the methods. There were no descriptions of the plant growth condition.
- The details of plant growth conditions are added in the revised manuscript (Lines 94-98).
There was no information about the metabolite identification and annotation in untargeted metabolomics using UPLC-TOF-MS analysis.
- The metabolite identification method is added in the revised manuscript (Lines 248-257).
Please describe the parameters for metabolite annotation according to the guideline by Fernie et al. 2011 (Plant Cell 23, 2477-2482). We cannot evaluate the correlation between metabolites and antioxidant activities in this study.
- The metabolite identification method is added in the revised manuscript (Lines 248-257).
The authors perform the Pearson correlation analysis based on Heatmap analysis between candidate metabolites and antioxidant activities. There is a widespread trade in the scientific research field to evaluate the correlation’s between total/individual metabolites with antioxidant activities of the sample of interest. However, the authors are not clear about the comments mentioned above of “We cannot evaluate the correlation between metabolites and antioxidant activities”.
- A few references among several are given below where the correlation between antioxidants with metabolite contents are evaluated:
- Park et al. 2018 (PLoS ONE 13(6): e0198739).
- Xu et al 2017 (Food Chemistry, 215, 149-156).
- Martino et al. 2013 (LWT - Food Science and Technology, 53, 327-330)
- You et al. 2012 (Journal of Chromatography A, 1240, 96-103).
- Sandhu et al. 2010 (Journal of Agricultural and Food Chemistry, 58, 4681–92).
- Kim et al. 2011 (Bioscience, Biotechnology, and Biochemistry 58, 4681–92).
- Kim et al. 2011 (Bioscience, Biotechnology, and Biochemistry 75 (4), 732–739).
L109 – TPC and TFC, the authors should explain each of your abbreviations the first time it appears in the main text.
- The complete form of TPC and TFC is added, and all other abbreviations are explained in the revised manuscript (Lines 101-102).
Statistical data analysis - You should describe PCA/PLS-DA in detail, including data transform and scaling. Regarding MetaboAnalyst, the version and the parameters used are also important to reproduce the data analysis.
- The details of multivariate analysis are described in the revised manuscript (Lines 222-242).
Figures and Tables
Fig. 4 – The resolution of this figure is too low. The font size is also small. The authors should describe all the colors/symbols (circles, triangles, and plus).
Fig. 5 – The authors should describe red and blue color values. For example, what does the value -2 mean? You mentioned “Data represent mean values ± SD (n = 3)” in this figure legend (L502-503). What does it mean?
- Figs. 4 and 5 are revised as per the reviewer's suggestions.
- The red and blue colors in the plot describe high and low intensities, and the values range from -2 to +2. The higher the red color intensity (from 1 to +2 values), the higher the metabolite con-tents and antioxidant activities; in contrast, higher blue color intensity (from -1 to -2 values) represents lower metabolite contents and antioxidant activities. The details of color and values are given in Figure’s legend (Lines 427-430; 509-513; 520-523).
Conclusions
This section is too short. Basically, the authors have considered only UPLC-MS based untargeted metabolites for the study which does not account for all metabolite profiles of this species. How do we use the results of the present study for further QTL analysis and plant breeding?
- The conclusion is reconstructed to clarify this study and future perspectives (Lines 576-600).
Data availability
- No (raw) data are available. You should make your data available in a public database, such as Metabolomics Workbench (http://www.metabolomicsworkbench.org/) and MetaboLights (http://www.ebi.ac.uk/metabolights/).
The raw data submission in the public database website is under submission processing. As it is time-consuming if the reviewer agrees, the authors can provide the raw data as the supplementary file to make it an online version of this article.
Reviewer 3 Report
Dear Editor,
I carefully read the manuscript by Darwish et al. that is well written.
My comments and suggestions for the authors are the following:
- Table 1 - Why did authors choose to indicate that a compoun was identified using the "Y"? Would not it have been clearer to use "YES"?
- Figure 4 - The legend of panel A is too small to be read. Authors should revise it accordingly.
- Authors should more deeply disclose the limitations of their study in the manuscript. In general, they included in the "Conclusions" paragraph pieces of information that should be more appropriately included in the Discussion. This aspect should be highly considered during revision.
Author Response
I carefully read the manuscript by Darwish et al. that is well written.
We thank the Reviewer for the time invested and the numerous comments that helped to improve our manuscript. We appreciate the Reviewer’s overall feedback on the importance of the scientific objective addressed in the present study.
My comments and suggestions for the authors are the following:
Table 1 - Why did the authors choose to indicate that a compound was identified using the "Y"? Would not it have been clearer to use "YES"?
- In Table 1, “Y” is changed to “YES” as per the reviewer's suggestion.
Figure 4 - The legend of panel A is too small to be read. Authors should revise it accordingly.
- Fig. 4 is revised as per the reviewer's suggestion.
Authors should more deeply disclose the limitations of their study in the manuscript. In general, they included in the "Conclusions" paragraph pieces of information that should be more appropriately included in the Discussion. This aspect should be highly considered during revision.
- The authors reconstructed the conclusion in the revised manuscript (Lines 576-591).
Round 2
Reviewer 2 Report
While I appreciate your efforts to address the concerns expressed during
the previous round of reviews, you failed to respond to my comment regarding metabolite identification as follows:
> Please describe the parameters for metabolite annotation according to the guideline by Fernie et al. 2011 (Plant Cell 23, 2477-2482). We cannot evaluate the correlation between metabolites and antioxidant activities in this study.
Secondly, regarding metabolite identification, the authors should cite the latest or original paper of all the databases/tools used. For example, the MassBank was presented by Horai et a. 2010. URL is not enough.
Furthermore, the authors should make your data available in a public database, such as Metabolomics Workbench (http://www.metabolomicsworkbench.org/) and MetaboLights (http://www.ebi.ac.uk/metabolights/).
I think that these considerations will make your work stronger.
